# Transcriptional diversification in a human-adapting zoonotic pathogen drives niche-specific evolution

Soma Ghosh [1,7], Chao-Jung Wu[1,4,7], Abraham G. Moller[1], Adrien Launay[1,5], Laina N. Hall [2,6], Bryan T. Hansen[2], Elizabeth R. Fischer[2], Jung-Ho Youn[3], Pavel P. Khil [1,3] & John P. Dekker [1,3] ✉

Bacterial pathogens can undergo striking adaptive evolutionary change in the context of infection, driven by selection forces associated with host defenses and antibiotic treatment. In this work, we analyze the transcriptional landscape associated with adaptation in an emerging zoonotic pathogen, *Bordetella hinzii*, as it evolved during a 45-month infection in an IL12Rβ1-deficient immunocompromised host. We find evidence of multiple niche-specific modifications in the intravascular and gastrointestinal compartments, involving the superoxide dismutase system, glutamate and ectoine metabolism, chaperone-mediated protein folding, pilus organization, and peptide transport. Individual blood lineages displayed modifications in glutathione, phenylacetate, and 3-phenylpropionate metabolism, iron cluster assembly, and electron transport, whereas individual gastrointestinal lineages demonstrated changes relating to gluconeogenesis, de novo pyrimidine synthesis, and transport of peptides and phosphate ions. Down regulation of the flagellar operon with corresponding loss of flagellar structures occurred in multiple lineages, suggesting an evolutionary tradeoff between motility and host immune evasion. Finally, methylome analysis demonstrates alteration of global genome methylation associated with loss of a Type III methyltransferase. Our findings reveal striking plasticity in how pathogen transcriptomes explore functional space as they evolve in the context of host infection, and demonstrate that such analysis may uncover phenotypic adaptations not apparent from genomic analysis alone.

Much work over the past decade has focused on how pathogens evolve in the context of colonization and infection in human hosts[1–5]. A variety of mutations associated with host adaptation and the evolution of antimicrobial resistance have been identified through genomic analysis of clinical isolates. Less well understood is how many of the mutations identified in these studies alter gene expression, and while transcription has been studied extensively in lab strains of bacteria, remarkably little is known about transcriptomes in clinical isolates and

[1]Bacterial Pathogenesis and Antimicrobial Resistance Section, Laboratory of Clinical Immunology and Microbiology, National Institute of Allergy and Infectious Diseases, National Institutes of Health, Bethesda, MD, USA. [2]Research Technologies Branch, Rocky Mountain Laboratories, National Institute of Allergy and Infectious Diseases, National Institutes of Health, Hamilton, MT, USA. [3]National Institutes of Health Clinical Center, National Institutes of Health, Bethesda, MD, USA. [4]Present address: School of Medical Laboratory Science and Biotechnology, College of Medical Science and Technology, Taipei Medical University, Taipei 110301, Taiwan. [5]Present address: Endogenomiks, Zapopan, Jalisco, Mexico. [6]Present address: University of California Berkeley, Berkeley, CA, USA. [7]These authors contributed equally: Soma Ghosh, Chao-Jung Wu. ✉e-mail: john.dekker@nih.gov

how they evolve under selection in the context of infection. Furthermore, inferring adaptive biochemical and physiologic changes from comparative genomic analysis of serially collected isolates becomes intractable in the background of extensive neutral mutagenesis, and here transcriptome analysis may reveal significant functional consequences that are not obvious from analysis of the underlying genomic changes alone.

In addition to genomic mutations, epigenetic changes may globally modify transcription, in particular changes in DNA methylation patterns[6]. In bacteria, DNA methylation occurs through the actions of DNA methyltransferases, which catalyze the transfer of methyl groups to specific positions on adenine (6 mA) and cytosine (4mC and 5mC) bases[7]. A great variety of DNA methyltransferases with diverse sequence specificities have been discovered, many in association with restriction modification systems. Individual methylation sequence motifs are relatively short and may occur thousands of times in a genome. Consequently, the action of single DNA methyltransferases can alter global genome methylation states, with the potential for generating large scale changes in gene expression patterns. Very little is known about how mutations in DNA methyltransferases modify transcription during host adaptation.

In this work, we explore intra-host transcriptional and epigenetic evolution in *Bordetella hinzii* that occurred during a previously characterized long term infection in an immunocompromised patient[8]. *B. hinzii* is a member of betaproteobacteria that shares genetic ancestry with the human-restricted pathogen *Bordetella pertussis*[9], but has been primarily associated with the respiratory tracts of poultry and rodent populations[10,11]. More recently it has been appreciated that zoonotic transmission of *B. hinzii* to humans causes a spectrum of infections including bacteremia, endocarditis, pneumonia, meningitis, and gastrointestinal tract disease with pancreatic involvement[8,11–17].

The clinical case on which this work is based involved persistent *B. hinzii* infection in a patient with a genetic IL-12Rβ1 deficiency (homozygous for IL12RB1 c.94 C > T p.Gln32Ter) seen at the NIH Clinical Center. Over a period of 45 months, 24 *B. hinzii* isolates were recovered from blood and stool cultures. Genomic analysis identified an E9G substitution within DnaQ, the proofreading ε-subunit of DNA polymerase III in 20/24 isolates, which increased the mutation rate by up to 1000-fold[8]. Hypermutation due to compromised DNA repair systems has been demonstrated previously to accelerate both host adaptation and the evolution of antimicrobial resistance in human pathogens[18–23]. However, little is known about the global transcriptional consequences of hypermutation.

To study genome-wide changes in gene expression that occurred during the course of host adaptation, we performed comprehensive transcriptome analysis and find evidence of remarkable transcriptional diversification in lineages with both intact and disrupted DNA proofreading, as well as apparent niche specific adaptations to the intravascular and gastrointestinal compartments. We also find evidence of a potential evolutionary tradeoff involving the down regulation of the flagellar operon and striking alteration of global genome methylation in one compound hypermutator lineage. This work demonstrates that the study of pathogen transcriptomes can supplement genomic analysis and shine light on the fundamental underlying biology of host adaptation.

## Results

### Large scale transcriptome sequencing and analysis in *B. hinzii*
To characterize the structure of the *B. hinzii* transcriptome and modifications that occurred over the course of 45 months of infection, we performed RNA-seq analysis on serial clinical isolates ($n = 22$; Figs. 1a, b) collected from our previous study[8]. Given the variability in growth rates among the 22 clinical isolates, we undertook RNA isolation at two distinct time points, 4 hours and 10 hours post-inoculation, thereby ensuring the capture of the exponential growth phase for all isolates. A comprehensive initial set of RNA-seq libraries ($n = 153$) was then generated and sequenced with Illumina technology (Supplementary Data 1). Clinical isolate 2B3 (GenBank: CP052845.1) was selected as the reference genome for read mapping, as it was deemed the closest intact representative to the inferred ancestral infecting genome and did not contain the large deletion present in isolate 1G1 (Figure S1a)[8]. The acquired sequencing reads were aligned to the 2B3 genome using BWA[24] and reads that mapped to annotated coding sequences ($n = 1.3–9.7$ million non-rRNA mapping reads per library) were enumerated using HTSeq[25]. Comparisons were performed using DESeq2[26] to identify differentially expressed genes (DEGs) across different groups of isolates.

Given the variable growth rates (Figures S1b, c), we focused our analysis on growth phase independent (GP-independent) genes to mitigate potential artifacts associated with asynchronous sampling of growth phase dependent (GP-dependent) genes in cross-isolate comparisons. To identify GP-dependent genes, we calculated the fold changes in gene expression between the 4-hour and 10-hour time points in the reference isolate 2B3. Genes displaying $\log_2$-fold changes ($\log_2$FC) that were 1 standard deviation (SD) greater or less than the mean were classified as GP-dependent, and this set of genes was subsequently excluded from downstream analyzes in all isolates (Supplementary Data 2 and Figure S2a). This criterion yielded a total of 3375 GP-independent genes, or 73.6% of the total. Principal Component Analysis demonstrated temporal clustering within the GP-dependent gene set that was largely absent from the GP-independent set (Figures S2b–e). A comparison of rlog counts between the GP-dependent and GP-independent sets to check for bias did not reveal systematic differences in expression levels (Figures S3a, b). To validate further the GP-independent gene set, we generated heatmaps enumerating the count differences between the 4-hour and 10-hour time points for both GP-dependent and GP-independent genes across all 22 clinical isolates (Supplementary Data 3–4 and Figures S3c, d). The resulting heatmap revealed a clear pattern in count differences for GP-dependent genes in all isolates compared to the GP-independent genes. We then utilized t-distributed stochastic neighbor embedding (t-SNE) plots to investigate the temporal separation of transcriptomes based on GP-dependent genes. These analyzes revealed separation of the 4 hr and 10 hr data points for GP-dependent genes (Figure S4a). In contrast, the 4 hr and 10 hr data points for the GP-independent genes did not exhibit appreciable separation (Figure S4b). These findings further support confidence that the effects of asynchronous sampling along the growth curves have been mitigated in this analysis. However, as noted in the sections below, there is a residual degree of weak growth phase dependence among the GP-independent gene set that is apparent as differences in the 4-hour and 10-hour transcriptomes in some analyzes that does not alter conclusions. Where relevant, we include the 4-hour analysis in the main text figures and the 10-hour analysis in the supplemental information.

### Serial clinical isolates exhibit striking transcriptional diversity uncorrelated with global genomic similarity
To investigate the extent of transcriptional diversity among the isolates, we employed relative Euclidean distance calculations based on rlog normalized counts of GP-independent genes at both 4-hour and 10-hour time points (Fig. 1c, Supplementary Data 3, Supplementary Data 5, and Figures S5a–c). Our analysis revealed substantial transcriptional diversity across the isolates, with Euclidean distances ranging from 15.9 to 163.1 (arbitrary units). At the 4 hr time point, the mean Euclidean distance between isolate transcriptomes was $58.3 \pm 18.8$, while at the 10 hr time point, the mean distance was $67.7 \pm 29.8$. To evaluate transcriptome similarity among the isolates and to compare with corresponding genomic similarity, we constructed unrooted dendrograms using the Euclidean distance matrix, thereby defining relationships between isolates in the space of

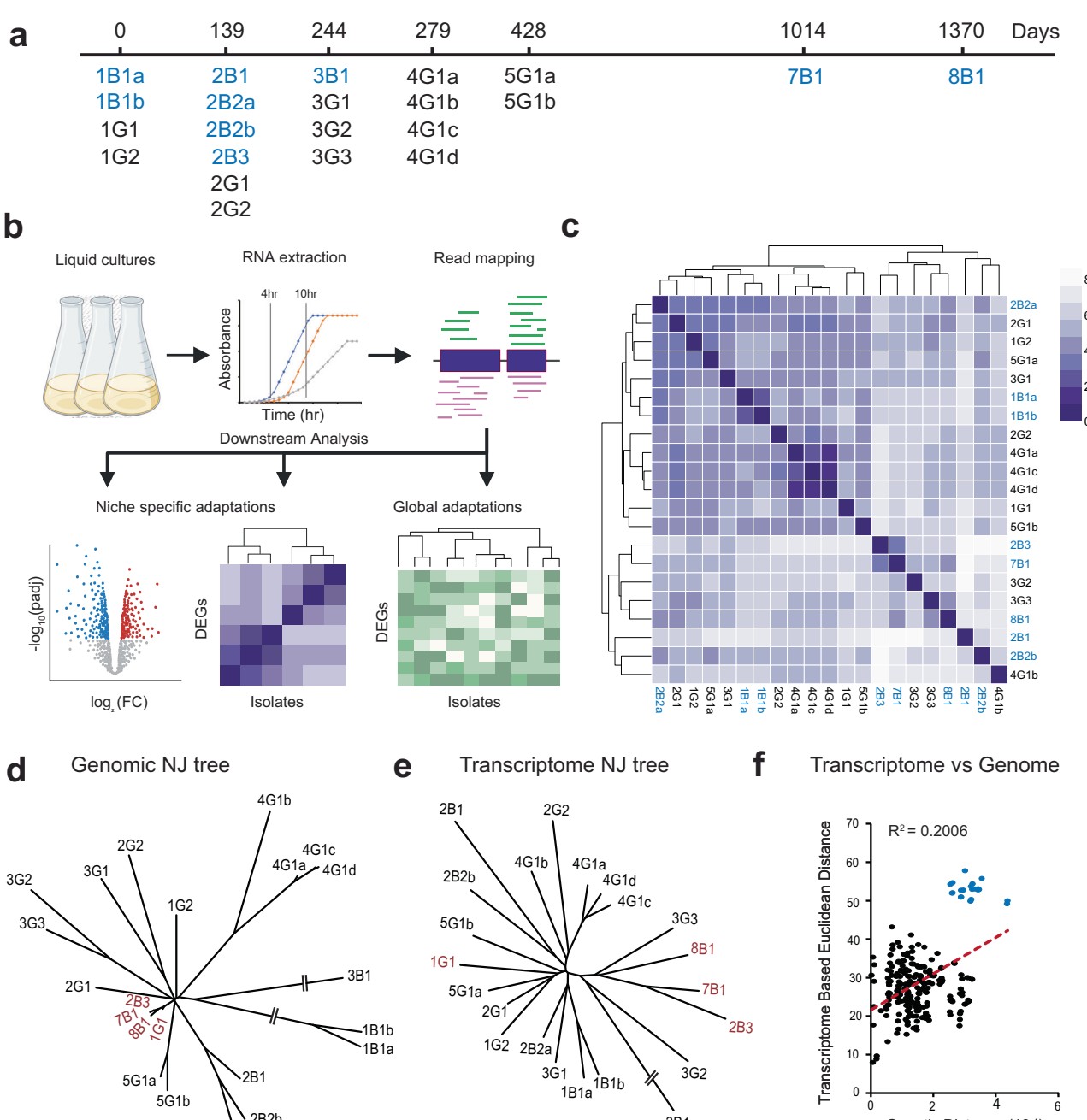

**Fig. 1 | *B. hinzii* isolates evolved extensive transcriptomic diversity during prolonged infection of human host. a** Timeline of isolate collection. Isolates were numbered by the day of collection, followed by culture source (Blood or Gastrointestinal), followed by number and letter designating individual morphotypes[8]. Blood isolates are indicated in blue and GI isolates are indicated in black. **b** Schematic illustration of experimental workflow for transcriptome analysis. Twenty-two *B. hinzii* isolates were cultured overnight in triplicate, followed by sub-culturing, RNA harvest at 4 hr and 10 hr and Illumina library preparation. The sequencing reads were trimmed, filtered, and mapped to reference isolate 2B3, followed by comparison of normalized counts of growth phase independent genes to identify differentially expressed genes using DESeq2. Created in BioRender. Ghosh, S. (2025) https://BioRender.com/r06y208. **c** Clustered heat map of relative transcriptome similarity (Fig. 1d, e and Figure S5d). Interestingly, this analysis revealed a tree that had a topology that was appreciably different from the genomic dendrogram in both the 4-hour and 10-hour calculations, suggesting that genomic relationships may not be Euclidean distances between the transcriptomes (only growth phase independent genes) of all isolate pairs, except 3B1 at the 4 hr timepoint. The color of each cell corresponds to the Euclidean distance between isolate pairs on a scale of 0–80, represented in the color band at right. **d** Neighbor joining tree based on the pair-wise distances between isolate genomes. **e** Neighbor joining tree based on Euclidean distances between isolate transcriptomes at the 4 hr timepoint. **f** Correlation plot between the genotypic distance and transcriptome distance (Euclidean distance) between pairs of isolates at 4 hr time point. DnaQ WT isolates are shown in red; isolates in black are DnaQ E9G proof-reading deficient hypermutators, and the isolate shown with blue dots is 3B1, which is a compound hypermutator and an outlier. Linear fit is to all points including 3B1.

transcriptome similarity (Fig. 1d, e and Figure S5d). Interestingly, this analysis revealed a tree that had a topology that was appreciably different from the genomic dendrogram in both the 4-hour and 10-hour calculations, suggesting that genomic relationships may not be predictive of transcriptome relationships among the isolates. This conclusion is reinforced by a quantitative comparison of the pairwise genome and transcriptome distances (Fig. 1f and Figures S5e–g), which indicates genomic similarity was not predictive of transcriptome

proximity in either the 4-hour or 10-hour datasets (4 hour $R^2 = 0.2$; 10 hour $R^2 = 0.02$), with the weak correlation that was present due to single outlier isolates 3B1 (compound hypermutator) in the 4 hour dataset. These observations suggest that in some cases individual mutations may lead to large changes in global patterns of gene expression, and in other cases the transcriptome may be remarkably stable to large numbers of mutations.

## Multiple niche-specific adaptations occurred in the transcriptomes of intravascular and gastrointestinal lineages

We next compared *B. hinzii* isolates from blood and the GI tract to probe for gene expression changes that might give insights into niche-specific biological adaptations (Figure S6a). A gene was considered differentially expressed if it had a log$_2$FC value ≥ 1 or log$_2$FC value ≤ −1 with an adjusted *p*-value of < 0.05 as calculated using DESeq2. A total of 268 genes exhibited differential expression between the blood and GI isolates, with 204 genes demonstrating relative downregulation in the GI tract isolates compared to 64 genes demonstrating relative upregulation (Supplementary Data 6). This included a genomic segment of ~60 kilobases (Kb) that was identified by the phage annotation tool PHASTER[27] as a putative phage from the Myoviridae family, related to phiEt88. Further examination of this genomic region in each isolate confirmed variable expression patterns relative to the reference isolate, 2B3 (Figure S6b).

Gene enrichment analysis of the 268 differentially expressed genes using topGO identified 9 biological processes that were significantly enriched in either blood or GI isolates ($P < 0.05$, Fisher's test) (Figure S6c). Processes upregulated in the GI lineages included those related to ectoine metabolism, pilus organization, peptide transport, and chaperone-mediated protein folding. In contrast, genes involved in glutamate biosynthesis, flagellar assembly, and protein secretion were downregulated in the GI isolates. These changes in gene expression correlated with enrichment in specific molecular functions potentially relevant to niche adaptation (Figure S6d). Relative upregulation of the superoxide dismutase system in blood isolates suggests that reactive oxygen species may have been a significant selection factor in the intravascular compartment, while relative upregulation of the ectoine system in GI isolates suggests that osmotic stresses may have played a role in the GI tract[28,29]. A third upregulated functional class in GI isolates was fimbrial usher porins involved in pilus assembly and translocation mediating attachment to host cells[30,31]. This finding suggests that the requirements of mucosal colonization shaped GI track adaptation, as has been demonstrated in other systems[32–34].

Biochemical pathway analysis of individual metabolic genes that were differentially expressed revealed insight into detailed metabolic reprogramming that occurred during niche adaptation (Fig. 2). Alterations broadly affected components of the tricarboxylic acid cycle, nitrogen assimilation pathways, and amino acid metabolism. Enzymes controlling precursor synthesis central to glutamate and aspartate metabolism were differentially expressed, as well as enzymes controlling pre-translational transamidation of Glu-tRNA$^{Gln}$ and Asp-tRNA$^{Asn}$. L-cystathione conversion into homocysteine and catabolism of L-cysteine to pyruvate was differentially regulated as well as phenylalanine to phenylacetate. In total this suggests differential programming of nitrogen handling and central oxidative phosphorylation in the GI and intravascular environments.

## Downregulation of flagellar assembly suggests evolutionary tradeoff between motility and immune escape

Comparison of niche-specific differential expression further revealed that genes involved in flagellar biosynthesis exhibited substantial transcriptional changes. Of the 204 genes that were downregulated in the GI lineages relative to the blood lineages, 19.6% were associated with flagellar biosynthesis[35–38]. Evaluation of the expression profiles of flagellar biosynthesis genes in individual isolates compared to the reference isolate, 2B3 (Figs. 3a, b and Supplementary Data 7) revealed that 9 of 13 GI isolates displayed an average of at least 32-fold downregulation in flagellar biosynthesis genes, with some genes downregulated by as much as 512-fold. Blood isolates 1B1a, 1B1b, 2B1, 2B2a, and 2B2b also demonstrated flagellar downregulation.

It was also observed that isolates with downregulated flagellar genes exhibited increased expression of genes involved in capsule biosynthesis (Fig. 3c and Supplementary Data 7). An inverse relationship between flagellar and capsular gene expression is present in Enterobacterales associated with Rcs phosphorelay signaling[39,40], and has also been described in *B. pertussis* in strains lacking a functional copy of the *risA* gene[41–43]. The *B. hinzii* isolates appear to have intact copies of *risA* as well as the gene encoding its signaling parter, *risS*, and both of these genes are expressed. Though it remains to be determined whether signaling is functional, it seems likely based on the above observations that counter-regulatory systems controlling flagellar and capsule biosynthesis are operative in *B. hinzii*, and that they were correlatively adjusted during host adaptation.

To correlate changes in flagellar gene expression with flagellar production, we experimentally quantified flagellar loss in a subset of isolates using transmission electron microscopy (TEM) negative staining (Figs. 3d, e). For isolates 7B1 and 2B3 with full flagellar gene expression, ~60% to 80% of bacteria demonstrated flagella. In contrast, for isolates 1G1, 5G1b, and 2G1, which exhibited significant downregulation of flagellar genes, less than 5% of the bacteria demonstrated flagella, confirming that decreased gene expression results in decreased flagellar production.

The findings of downregulation of flagellar synthesis in different lineages argues for selection and suggests the possibility that an evolutionary trade-off between motility, conferred by flagella, and the antigenic liability posed by these large and exposed structures may have occurred during the course of infection.

## Global modification of the methylome occurred in a compound hypermutator lineage in the intravascular space

Given the role of DNA methylation in shaping bacterial gene expression[6], we evaluated the methylome of the clinical isolates to discern changes in methylation patterns during the course of infection. We utilized nanopore technology to sequence the native genomic DNA (gDNA) of the 22 clinical isolates in conjunction with PCR-amplified (methylation free) gDNA for reference isolate, 2B3. By comparing the profiles between these two libraries using Nanodisco[44,45], we identified two methylated motifs (Supplementary Data 8). A Type II motif involving a 5mC modification in a palindromic sequence (CGC5mCGGCG) was found to be present in all isolates. A second motif involving a 4mC modification in a Type III sequence (AGCG4mCCY) was found in all isolates except 1B1a and 1B1b, suggesting the possibility that modification of the methylome occurred during the course of intra-host evolution (Fig. 4a).

Analysis using DNA Methylase Finder[45] identified 7 potential methyltransferases in the clinical isolate genomes (Fig. 4b) and revealed that an annotated Type III DNA methyltransferase gene (HHA24_19560) harbored a stop gain mutation only in isolates 1B1a and 1B1b (Fig. 4c). Given the exact correspondence between the stop gain mutation in the Type III DNA methyltransferase shared by 1B1a and 1B1b and the loss of the Type III methylated motif in these same isolates, we concluded that it is highly probable this enzyme is responsible for methylation at this position, and that the stop gain mutation was responsible for its loss. It is worth noting that the Type III motif occurs 4479 times within the positive and negative strands of the genome (Fig. 4d). Consequently, the loss of methylation at this sequence results in a global alteration in methylation across the genome, which may have substantial consequences for transcriptional regulation[6].

Isolates 1B1a and 1B1b are notable for being compound hypermutators, containing both the DnaQ E9G substitution disabling DNA

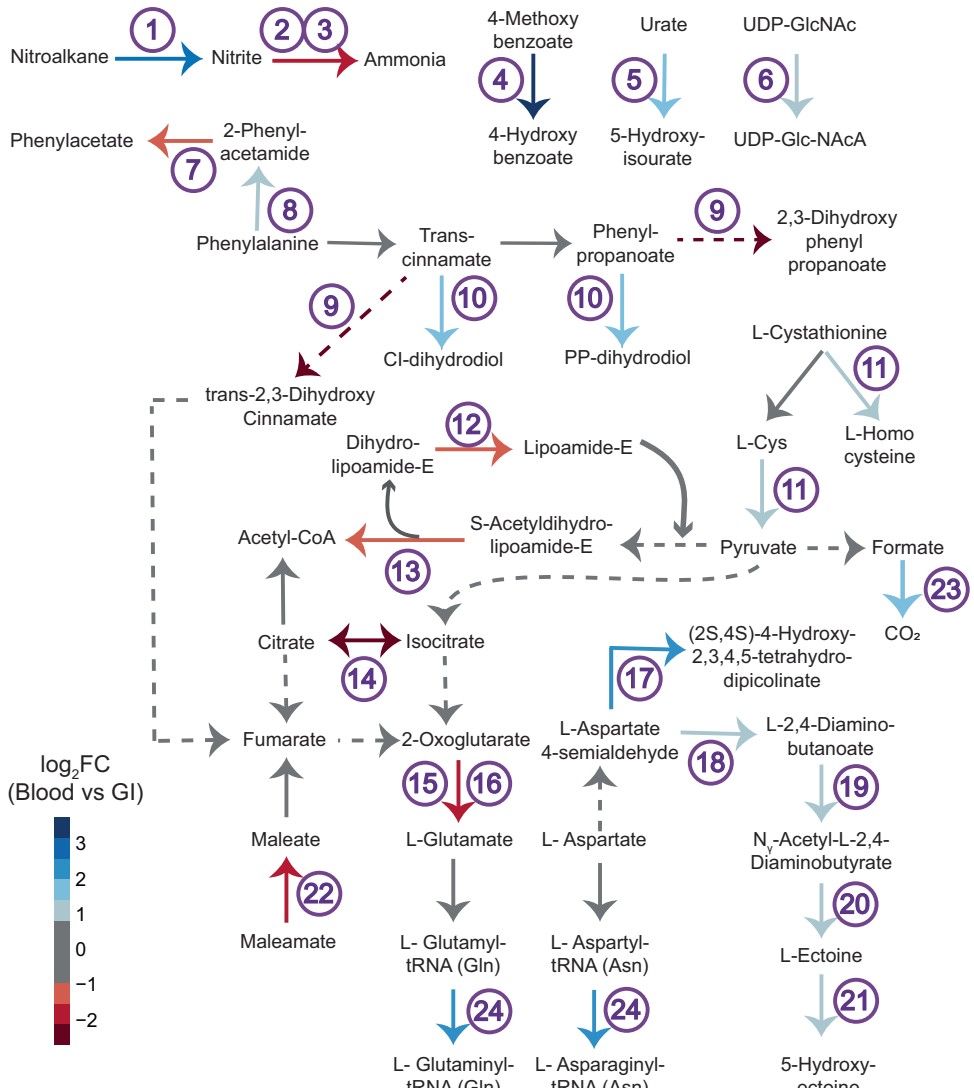

**Fig. 2 | Niche-specific differential regulation of central metabolic pathways.** A total of 268 differentially expressed genes were identified between isolates harvested from blood and the GI tract, of which 25 corresponded to metabolic genes in the KEGG database. The figure indicates the enzyme, intermediates, and products for each inferred reaction, and enzymes are numbered as defined below. Arrow color defined in key indicates the log₂FC values of differential expression based on group DESeq2 calculations. Enzyme labels are as follows: 1) nitronate mono-oxygenase, 2) nitrite reductase (NADH) large subunit, 3) nitrite reductase (NADH) small subunit, 4) 4-methoxybenzoate monooxygenase, 5) dihydropyrimidine dehydrogenase subunit A, 6) UDP-N-acetyl-D-glucosamine 6-dehydrogenase, 7) acylamidase, 8) catalase-peroxidase, 9) 3-hydroxycinnamic acid hydroxylase, 10) rhodocoxin reductase, 11) cystathionine beta-lyase, 12) dihydrolipoyl dehydrogenase 13) pyruvate dehydrogenase E2 component, 14) aconitate hydratase, 15) glutamate synthase (NADPH) large chain, 16) glutamate synthase (NADPH) small chain, 17) 4-hydroxy-tetrahydrodipicolinate synthase, 18) diaminobutyrate–2-oxo-glutarate transaminase, 19) L-2,4-diaminobutyric acid acetyltransferase, 20) L-ectoine synthase, 21) ectoine hydroxylase, 22) maleamate amidohydrolase, 23) NADH-dependent formate dehydrogenase delta subunit FdsD, 24) aspartyl-tRNA(Asn)/glutamyl-tRNA(Gln) amidotransferase subunit A.

mismatch repair and a second substitution in the MutM protein of the base excision repair pathway. As a result, these isolates accumulated a substantial number of mutations during the infection ($n = 1055$ for 1B1a and $n = 1042$ for 1B1b relative to the reconstructed ancestor)[8], likely contributing to the observed degree of perturbation of transcription (Fig. 4d, Supplementary Data 9). Comparing the transcriptomes of all isolates relative to 2B3 revealed that 1B1a and 1B1b contained a set of 33 genes with greater than 2-fold changes in expression (Fig. 4d) that were either absent or present in the opposite direction in all other isolates, representing potential candidates for methylation-dependent changes. Seven of these genes contained at least one AGCG4mCCY motif (Fig. 4d) located within 500 bases and one contained this motif within 100 bases in these two isolates. It should be noted that expression of genes is likely altered by methylation at more distant positions through the secondary action of a variety of regulatory

systems, such that the adjacency analysis above may not be predictive of such effects.

## Clinical lineages demonstrate unique evolutionary trajectories with both shared and distinct functional adaptations

We next sought a global analysis of the entire set of clinical transcriptomes sampled throughout the infection to identify overall patterns of potential functional adaptation. We compared the DEGs in each isolate relative to the reference isolate 2B3, chosen because it represents the isolate closest to the inferred common ancestor with an intact genome, and hence its transcriptome may be expected to be among the most representative of the original infecting lineage, with changes relative to this transcriptome representing modifications that occurred during the course of intra-host evolution (Supplementary Data 7, Figure S7). This analysis revealed a number of genes that

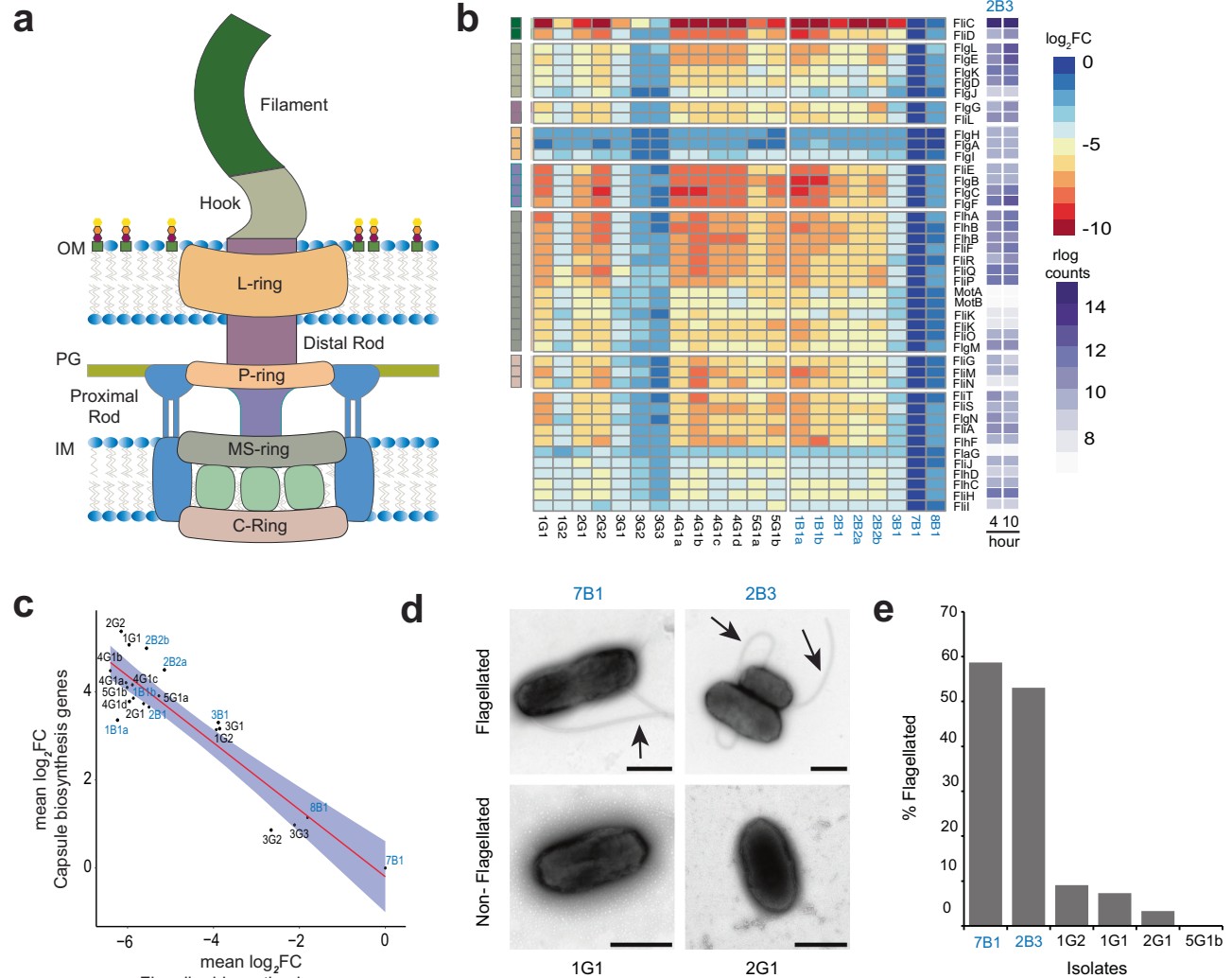

**Fig. 3 | Downregulation of flagellar biosynthesis suggests evolutionary trade-off. a** Schematic representation of the flagella with key components labeled and color coded. **b** Heatmap of flagellar genes (rows) vs. isolates (columns). Data are aggregated by isolate and colors correspond to log₂FC of flagellar genes expression relative to ancestral reference isolate 2B3. A separate heatmap represents the absolute rlog count values of the genes in 2B3 at the 4 hr and 10 hr time points. **c** Correlation plot between the mean of log₂FC for flagellar genes and genes involved in capsule biosynthesis in all isolates relative to ancestral 2B3 reference.

Data are fit with linear model and envelope represents the 95th percentile confidence interval of the linear fit. **d** Representative TEM images of select isolates, following overnight growth. Each image was selected as a field from a single biological TEM preparation of the represented isolate. (see "Methods"). Flagella are marked by black arrows. Scale bars are 500 nm. **e** TEM images for each isolate following overnight growth were scored for the presence of flagellated bacteria. A minimum of 30 bacteria were counted and % of flagellated bacterial cells plotted.

displayed differential regulation in small groups and individual isolates, in addition to the above discussed differential regulation of major pathways including flagellar and capsular biosynthesis. To look for broader features of functional adaptation, we performed pathway enrichment analysis of differentially expressed genes across the entire set of isolates with respect to 2B3. This yielded a total of 31 enriched GO categories that were differentially regulated across all isolates, with three of these shared by all isolates (with respect to 2B3) (Supplementary Data 10).

To visualize the relationships revealed by GO analysis, we constructed a bipartite network linking isolates and ontologies. In this analysis, isolates were segregated according to their compartment of origin and day of culture, and the ontologies were grouped based on whether they were restricted to isolates derived from one source or were shared among isolates derived from both sources (Fig. 5). The analysis revealed that most of the source-specific GOs were present uniquely only in one isolate (11/16), and the remainder were present in

small groups (4/16 shared by two isolates and 1/16 shared by three isolates) that were phylogenetically and temporally proximate. This suggests that multiple independent niche-specific functional adaptations were discovered by different lineages throughout the course of the infection. The unique enriched categories in blood isolates included glutathione, phenylacetate, and 3-phenylpropionate metabolism, iron cluster assembly, and coupled electron transport. The unique enriched categories in the GI isolates included gluconeogenesis, de novo pyrimidine synthesis, peptide and phosphate ion transport, and response to oxidative stress.

GO categories that were shared by isolates from both compartments included cellulose, glutamine, glutamate, and UDP-glucose metabolism, as well as amino acid, organic phosphate, and xenobiotic transport. A minority of the shared enrichments demonstrated an apparent temporal sequence among the blood isolates which was largely absent from the GI isolates. Overall, however, the network graph does not reveal global evidence of temporally sequential

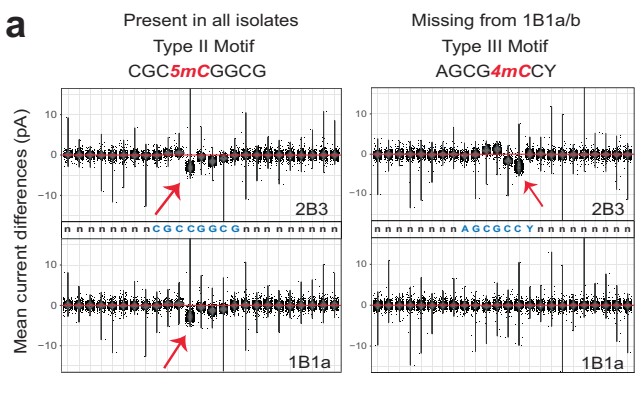

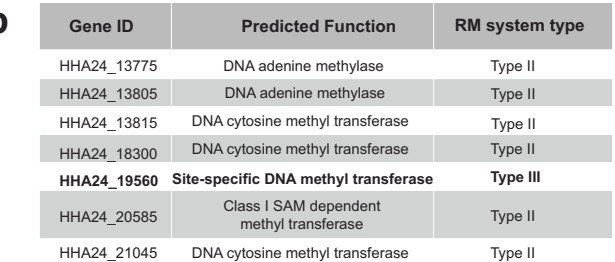

**Fig. 4 | Global loss of Type III genome methylation in compound hypermutator sublineage. a** The plot represents mean current differences (pA; y-axis) between the native methylated DNA and the PCR-amplified DNA as obtained from ONT Nanopore sequencing and analyzed using nanodisco[44]. Data from each isolate represents a single biological replicate and the display items represent the standard visual output of the nanodisco software, with violin plots summarizing the underlying distribution[44]. Red arrow highlights the difference in mean current levels identified by nanodisco to represent a specific DNA modification (see "methods"). Left panel represents current difference in a Type II motif (CGC5mCGGCG) and right panel represents current difference in a Type III motif (AGCG4mCCY) in isolates 2B3 (top) and 1B1a (bottom), demonstrating loss of the Type III motif in 1B1a. **b** Potential methyltransferases identified with MethylaseFinder[45]. **c** Multiple sequence alignment of putative Type III DNA methyl transferase shows a stop-gain mutation at position 179 in 1B1a and 1B1b, explaining loss of Type III methylation in these isolates. Alignment is shown for residues 160–190. **d** Circos plot summarizing the distribution of genes and methylation motifs in the 2B3 genome. Rings are described in the following from outermost to innermost. The outermost ring identifies genes in 1B1a and 1B1b whose transcription is potentially modified by loss of Type III motif methylation. Pink indicates the 33 gene (in 25 groups) that are differentially expressed in both 1B1a and 1B1b (relative to all other isolates with intact Type III methylation). The region is zoomed in 75% for better visualization. Dark green indicates those genes that are not differentially regulated (these genes are summarized and piled). The next two rings identify Type III methylation motif positions on the positive strand (purple) and negative strand (blue). The inner two rings represent $\log_2$FC of genes in 1B1a (next to innermost ring) and 1B1b (innermost ring) with reference to 2B3. Full data are given in Supplementary Data 6 and 8. Plot was generated using Circos package[76].

accumulation of functional category enrichments, largely consistent with the bundle-like phylogenetic structure of the isolates, indicating independent evolution of most lineages.

## Discussion

Bacterial pathogens are known to undergo remarkable genomic and phenotypic adaptive change in the context of host infection[1–5]. These modifications are driven in part by selective forces associated with host immune responses, clinical antibiotic treatments, and metabolic requirements to support invasion into new biological compartments. Evolution in response to these forces shapes virulence, antibiotic resistance, and the establishment of chronic infection. Analysis of serial clinical isolates from individual patients has revealed substantial underlying genomic plasticity in a variety of pathogens, in some cases accelerated by hypermutation due to lesions in DNA mismatch repair or proofreading processes, and the movement of active mobile genetic elements[1–5].

While much has been learned from detailed analysis of genomic mutations in these cases, the parallel study of global transcriptional changes has been relatively neglected. Characterization of how gene expression networks explore functional space during host adaptation can potentially reveal information about selection forces and responses that may not otherwise be apparent from genomic analysis alone. Our previously characterized case of an unusually prolonged infection

with *B. hinzii* in a patient with a genetic immunodeficiency provided a unique opportunity to examine intra-host evolution of the transcriptome in a set of well-characterized isolates spanning a 45-month clinical infection[8].

The isolates included a subset with hypermutation due to a substitution in the ε-proofreading subunit of DNA polymerase III (E9G in DnaQ) and lineages with additional secondary mismatch repair deficiencies due to substitutions in MutM and MutY. Comparison of these otherwise clonal lineages permitted evaluation of how hypermutation generates transcriptome diversity. Surprisingly, we found that genomic distance measured by pairwise substitutions was largely uncorrelated with transcriptome similarity measured by relative Euclidean distance. These observations suggest that small numbers of mutations, likely located within important regulatory regions, may lead to global changes in transcription, and conversely, the transcriptome can remain remarkably stable even with large numbers of mutations, perhaps due to selection against mutations in important regulatory regions.

Comparison of blood and GI lineages revealed that multiple potential niche-specific adaptations occurred. Processes related to hydrogen peroxide and ectoine metabolism, pilus organization, peptide transport, and chaperone-mediated protein folding were upregulated in GI isolates and those related to glutamate biosynthesis, flagellar assembly, and protein secretion were downregulated. The superoxide dismutase system, which functions to mitigate damage

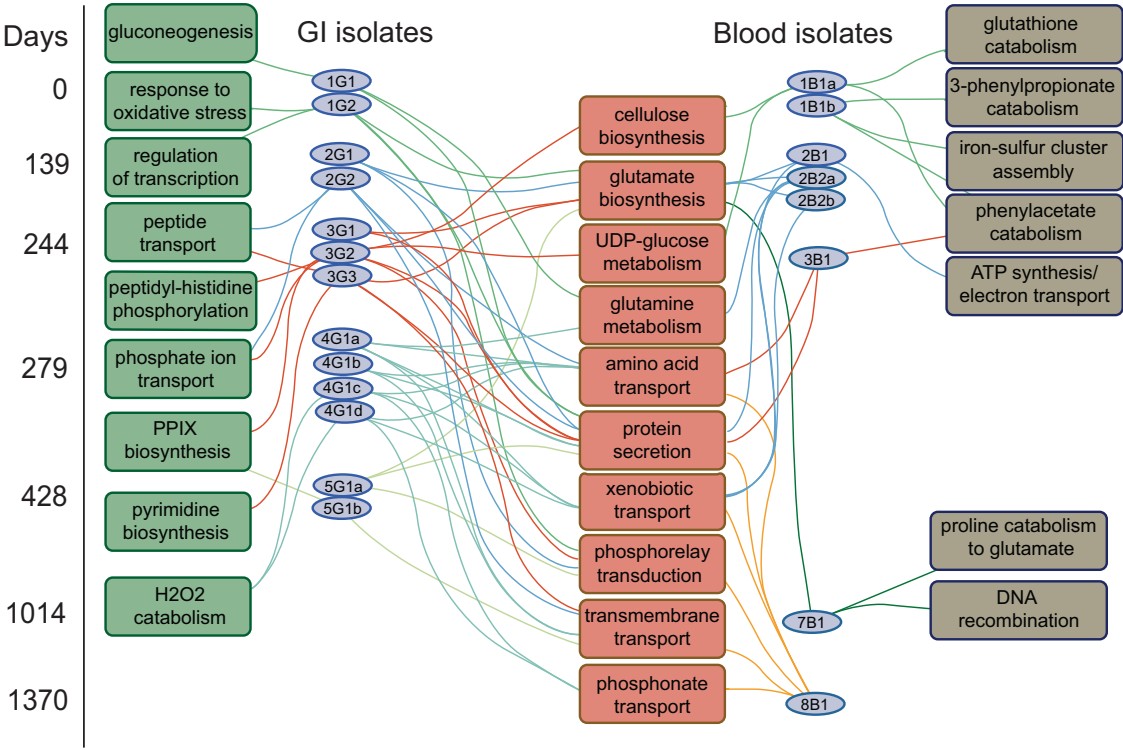

**Fig. 5 | Functional classification of global transcriptional changes reveals both unique and shared adaptations.** Bipartite network representing the relationships between isolates (ellipses) and enriched functional categories (rectangles) as determined by Gene Ontology classification of differentially expressed gene groups. The isolates are ordered by day of culture (left) and compartment (Blood, GI). Enriched functional categories uniquely present in the GI isolates are shown on the left, those that are uniquely present in blood isolates are show on the right, and those shared by isolates cultured from both blood and GI compartments are shown at the center. Three enriched functional categories shared by all isolates relative to 2B3 are not shown.

due to oxygen radicals[46] was upregulated in blood lineages, and indeed, our previous mutational spectrum analysis of these isolates identified a strong oxidative stress signature (excess of G:C > T:A transversions) reporting on oxidative conditions encountered in vivo during persistent host infection[8]. Another finding was increased expression of genes involved in the metabolism of ectoine and hydroxyectoine, multifunctional compatible osmolytes that confer resistance to osmotic, salt, and temperature stress. Ectoine and hydroxyectoine are produced by extremophiles and halophiles, and the acquisition of the ectoine biosynthesis cluster has been associated with habitat expansion into marine and other high or variable salinity environments[28,29,47]. Ectoine may play a protective role in the conditions encountered in the GI track where the osmolarity and salinity of luminal contents may fluctuate over wide ranges depending on host dietary content and relative stool hydration. A third upregulated functional class involved fimbrial usher porins. These outer membrane proteins are involved in pilus assembly and translocation of fimbrial polymers to bacterial cell surface that mediate attachment to host cells and tissues and host colonization[30,48,49]. The above findings suggest that osmotic stress, as well as the requirement for cellular adhesion, may have figured importantly into the fitness landscape seen by the GI isolates, while oxidative stress was an important selection force for the blood isolates.

The finding of downregulation of flagellar synthesis in different lineages argues for selection and points to an evolutionary trade-off between motility, conferred by flagella, and the antigenic liability posed by these large and exposed structures[50–54]. While the patient in this case had impaired IL-12Rβ1-dependent intracellular immunity, it is expected that components of innate pattern-recognition immunity including that mediated by TLR5 were likely intact to mediate such selection[55]. A tradeoff between motility and immune escape has been

suggested in pathogens including *E. coli* and *Salmonella Typhimurium* in which flagellar expression was reduced or eliminated in response to host flagellin-mediated immune-clearance during infection[50–54]. The exact molecular mechanisms of flagellar downregulation in the *B. hinzii* lineages, however, remain unclear, and mutational subset analysis failed to reveal simple associations between individual mutations or groups of mutations and flagellar phenotype. A Q304L substitution within the FliF flagellar M-ring protein of uncertain significance is shared among all DnaQ E9G lineage isolates. The majority of these isolates demonstrated strongly downregulated flagellar expression; however, isolates 1G2, 3G1, 3G2, and 3G3 contained FliF Q304L and did not demonstrate strong flagellar downregulation. Separately, isolate 1G1 of the DnaQ WT lineage demonstrated flagellar downregulation without FliF Q304L. Given the lack of an obvious group of mutations common to all isolates, it appears possible that downregulation evolved at least twice convergently with different mutations. Further experimental work will be required to determine the functional consequences of each of the observed mutations on flagellar expression to establish exact mechanisms involved.

We also observed a strong inverse relationship between flagellar biosynthesis and capsule production, with capsule biosynthesis strongly activated in the lineages that lost flagellar expression, as has been described previously in Enterobacterales in association with Rcs phosphorelay signaling[39,40] and in *B. pertussis* in strains lacking a functional copy of the *risA* gene[41–43]. The *B. hinzii* genomes studied here contain what appear to be intact copies of *risA* and the gene encoding its signaling partner, *risS*, and both genes are expressed in the isolates. Further experimental work will be required to determine the underlying mechanism controlling inverse expression, and whether the production of capsule is associated with increased fitness of these lineages.

The identification of changes in the genomic methylation profiles within a subset of isolates during host adaptation represents a significant and novel finding in this study. While genome methylation has been studied in eukaryotes for decades, only relatively recently have methods become available allowing large scale characterization of bacterial genome methylation, and these methods have not been widely applied to examine changes during host adaption[44,45]. DNA methylation is known to have the potential to modify transcription, sometimes dramatically[6]. Given that individual bacterial DNA methyltransferases often methylate thousands of motifs within a genome, the possibility exists for global changes in methylation patterns resulting from loss of single methyltransferases with consequent large-scale changes in transcription. Our finding of loss of methylation at a Type III motif represented more than 4000 times throughout the genome during the course of clinical infection demonstrates that these changes can indeed occur and opens intriguing possibilities for understanding the role of epigenetic regulation in bacterial adaptation during infection. Deeper experimental work will be required to characterize the precise transcriptional and phenotypic consequences of the loss of methylation found in isolates 1B1a and 1B1b.

The examined clinical lineages demonstrated unique evolutionary trajectories with both shared and distinct functional adaptations. Many individual enrichments in gene expression functional categories were present only in single isolates. Glutathione, phenylacetate, and 3-phenylpropionate metabolism, iron cluster assembly, and coupled electron transport were enriched in blood isolates, and gluconeogenesis and de novo pyrimidine synthesis were enriched in GI isolates. The findings will require deeper experimental study, but they suggest that a variety of global metabolic changes occurred during host adaptation.

An important caveat of this work is that transcriptomes were examined in standardized LB culture media. It is likely that there are transcriptional processes that are dynamically regulated by the native host environment that may have undergone specific selection and adaptation that were not appreciated in this study. Future work aiming at transcriptome characterization in native contexts, including the intracellular environment of phagocytes will be necessary to answer these questions.

In conclusion, our findings reveal surprising plasticity in how pathogen transcriptomes explore functional space as they evolve during the course of host infection. These underlying changes involved a variety of processes ranging from reprogramming of core metabolism, to responses to oxidative and osmotic stresses, to cell wall structure and host cell adherence. Evidence of an evolutionary trade-off between flagella and immune escape was present, and global changes in genome methylation occurred during the course of the infection. We believe these findings demonstrate that the study of intra-host evolution of pathogen transcriptomes and methylomes may uncover important phenotypic adaptations not otherwise obvious from genomic analysis alone, potentially revealing unappreciated dimensions of host-pathogen interactions and the associated selection forces that drive pathogen evolution.

## Methods

### Ethics statement

The research presented complies with all relevant ethical regulations. Informed written consent was obtained from the patient under National Institutes of Health (NIH) Institutional Review Board (IRB) protocol 93-I-0119 upon admission to the NIH Clinical Center. This protocol was approved by the NIH IRB committee. Diagnostic clinical cultures were performed as part of routine standard-of-care management under this consented protocol, and only de-identified sub-cultured bacterial isolates were used in the work presented in this manuscript. The results of the work in this manuscript were not used for patient care. The work presented in this manuscript was thereby excluded from further NIH IRB review, on the basis of the fact that it

was a study of a single case, involving only sequencing and analysis of de-identified bacterial isolates.

### Bacterial strains

The work in this study is based on 22 clinical isolates collected over the course of 45 months from the blood and stool specimens of a single patient at the NIH Clinical Center. These isolates were previously characterized, sequenced, and annotated in Launay et al.[8]. Additionally three *B. hinzii* strains (ATCC 51730, ATCC 51783 and ATCC 51784) were obtained from American Type Culture Collection (ATCC) to serve as outgroups for the construction of the phylogenetic tree.

### Growth curves

To obtain growth curves, isolates were grown in LB medium at 37 °C overnight, followed by sub-culture in fresh LB medium at OD600 = 0.01. Growth was then monitored in a plate reader for 36 hrs. For each isolate, an early exponential phase interval was identified, and a linear function was fit to the $\log_2$-transformed OD600 values in R. The slope of the fitted line was taken as a best estimate of growth rate, and the doubling time was taken as its inverse. In most cases the log-linear fit was excellent. In a couple cases, the slope appeared to have more than one component. The best single exponential log-linear fit was taken in these cases for the purposes of this analysis.

### RNA extraction and library preparation

Isolates were recovered from frozen stocks onto blood agar plates so that initial colonies could be visualized and confirmed prior to starting cultures for sequencing. Verified bacterial colonies from each plate were inoculated into 5 ml LB broth (Cat # 244620, Becton Dickinson) and incubated at 37 °C with 220 rpm shaking overnight. Following growth, cultures were adjusted to OD600 of 0.01 in 10 ml LB broth and incubated at 37 °C with 220 rpm shaking. At either 4 hr or 10 hr time points during incubation, individual cultures were treated with RNA-protect (Cat # 76506, Qiagen) following manufacturer's instructions. Total RNA was extracted using MagMAX Viral/Pathogen Nucleic Acid Isolation kit (Cat # A42352, ThermoFisher Scientific) implemented on a KingFisher Flex purification system (Cat # 5400610, ThermoFisher Scientific). Genomic DNA was removed with DNase treatment (Cat # AM2238, ThermoFisher Scientific). Following depletion of ribosomal RNA using the NEBNext rRNA Depletion kit (Cat # E7850, New England Biolabs), RNA-seq libraries were built using the NEBNext Ultra II Directional RNA Library Prep kit (Cat # E7760, New England Biolabs) and sequenced with an Illumina NextSeq 550 in either 150 bp paired-end mode or 75 bp single-end mode.

### Sequencing of ATCC reference isolates

DNA was extracted from the three ATCC isolates (ATCC 51730, ATCC 51783, and ATCC 51784) with the Maxwell HT 96 gDNA Blood Isolation kit (Cat # A2670, Promega) following the manufacturer's protocol. Extracted DNA was quantified and quality controlled using the methods described in the Methylation Motif Discovery section described below. Nanopore libraries were prepared using the rapid PCR barcoding kit (Cat # SQK-RPB004, Oxford Nanopore Technologies) using 7.5 ng genomic DNA as input and following manufacturer's instructions. Libraries were sequenced for 48 hours using a GridION instrument running MinKNOW 21.05.8. DNA sequences were base-called on the instrument itself using Guppy 5.0.11[56] set for "super accurate mode". Illumina libraries were prepared from extracted genomic DNA using Illumina DNA Prep (Cat # 20060060, Illumina) and Nextera DNA CD Indexes (Cat # 20018707, Illumina) and sequenced as 150 bp paired-end reads with a NextSeq 550 instrument. Short reads were adapter and quality trimmed using TrimGalore 0.6.7 using options –nextseq20 –paired[57].

For genome assembly, long reads were first assembled using flye 2.9.1[58] using default parameters. Each flye assembly was then polished

with corresponding Illumina reads using Pilon 1.23[59] with default parameters. The ATCC genomes were used to build the phylogenetic tree in Figure S1.

### Read mapping and differential expression analysis

For samples that were sequenced in 150 bp paired-end mode, cutadapt v2.10[57] was used to trim R1 reads, and only the first 75 bp of R1 was used for downstream analysis. Illumina adapters were removed, and low-quality reads (Phred score below 20) were filtered by TrimGalore v0.6.6[57]. Reads of patient isolates were aligned to isolate 2B3 (Gen-Bank: CP052845.1) with BWA v0.7.17[24]. Aligned reads were sorted with Samtools v1.13[60,61], and reads mapping to CDSs were counted by HTSeq v0.11.4[25] with parameter "stranded reverse" and mode "union". Genes with read counts <10 in all isolates were not included in the analysis.

To identify growth phase independent genes (see explanation in Results section), genes that were differentially expressed, defined as $\log_2$ fold-change ($\log_2$FC) greater than mean + 1 SD or less than mean - 1 SD, between the 4 hr and 10 hr time points in reference isolate 2B3 were categorized as growth phase dependent and were filtered from the data set. For the remaining 3375 growth phase independent genes, $\log_2$FC was calculated for different sets of isolates. DESeq2 analysis was performed with two designs. The comparison between blood and GI isolates used d = -hour + source. For the comparison of each isolate with reference to 2B3, design d = -hour + isolate was used, followed by comparison of individual isolates using the apeglm shrinkage estimator[62]. Genes with $\log_2$FC $\geq 1$ or $\leq -1$ and adjusted $p$-value < 0.05 after Benjamini-Hochberg correction were identified as differentially expressed.

### Functional annotation and gene enrichment analysis

To perform functional annotation, the entire set of PGAP-annotated 2B3 protein sequences was compared to the NCBI nr database using BLAST v2.10.0[63,64] and scanned in the InterPro database using InterProScan v5.42-78.0[65,66]. Blastp and InterProScan outputs were imported into blast2go basic[67] and merged prior to downstream analysis. GO term mapping and annotation were performed using blast2go basic and gene enrichment analysis were conducted using topGO v2.40.0[68] with weight01 algorithm and Fisher exact test.

### Methylation motif discovery

Nanopore sequencing signals of native and corresponding PCR amplified DNA were compared using nanodisco1.0.3_dev (Tourancheau, A., et al., Nature Methods, 2021). Genomic DNA was extracted from 22 clinical isolates with the Maxwell HT 96 gDNA Blood Isolation kit (Cat # A2670, Promega) following the manufacturer's protocol. Extracted DNA was quantified using Qubit dsDNA BR Assay kit (Cat # Q32850, Thermo Fisher Scientific) and analyzed for fragment integrity using a 4200 TapeStation System (Cat # G2991BA, Agilent) with DNA Reagents (Cat # 5067-5366, Agilent) and DNA ScreenTape (Cat # 5067-5365, Agilent). Native (methylated) DNA nanopore libraries were prepared for the 22 clinical isolates using the Rapid Sequencing Kit (Cat # SQK-RAD004, Oxford Nanopore Technologies) according to the manufacturer's protocol. Libraries for the PCR-amplified (non-methylated) genomic DNA were generated for 2B3 with the rapid PCR barcoding kit (Cat # SQK-RPB004, Oxford Nanopore Technologies) using 7.5 ng of genomic DNA as input and the PCR extension time set to 7 minutes 30 seconds. Libraries were sequenced for 48 hours on a GridION instrument (Oxford Nanopore Technologies) running Min-KNOW 21.05.8. DNA sequences were basecalled on the instrument itself using Guppy 5.0.11[56] set for "super accurate" mode.

FAST5 files were gzip compressed before beginning the pipeline using compress_fast5 4.0.0 from the ont_fast5_api run with default parameters. Reads were preprocessed using the 2B3 genome as the reference for all the clinical isolates using the nanodisco[44,69] "preprocess" command with default parameters. To facilitate parallel processing, the genome was divided into 300 genomic chunks of 5 kb each, covering 1.5 Mb genome. Differences in measured currents between the native and PCR-amplified sequences were determined for 100 groups of three chunks each using the nanodisco "difference" command with default parameters and then merged into a single difference file using the nanodisco "merge" command with default settings. Motif discovery was carried out until no new motifs were found, followed by manual refinement to optimize confidence scores as per instructions provided in the manual. Motifs with confidence scores of 50% or lower for the methylated base were discounted from further analysis. Two high confidence refined motifs were discovered (Supplementary Data 8) that were further characterized for methylation typing and fine mapping using the nanodisco "characterize" command. Motif positions on the 2B3 genome was determined using the SeqKit "locate" command on both strands[70].

### Electron microscopy imaging and quantification

Bacterial cultures grown overnight at 37 °C were harvested by centrifugation at 420 g for 10 min to obtain ~5 billion cells. Samples were washed twice in 1X PBS, followed by fixation in 4% paraformaldehyde. 5 µl droplets of concentrated fixed bacteria in suspension were allowed to settle for 30 s onto freshly glow-discharged carbon coated 200 mesh copper grids. Excess was wicked away with filter paper. 5 µl droplets of dH2O were applied for 30 s, excess removed by wicking with filter paper, then stained for 30 s with NanoVan (Ted Pella, Inc.) and an additional 30 s dH2O wash prior to final wicking. The grids were allowed to dry prior to examination at 80 kV on a Hitachi H-7800 (Hitachi High Technologies Corporation) with digital images acquired with an AMT XR81 camera (Advanced Microscopy Techniques, Woburn, MA) or 80 kV on a FEI BT Tecnai transmission electron microscope (Thermo Fisher/FEI, Hillsboro, OR) and digital images were acquired with a Gatan Rio9 camera (Gatan, Pleasanton, CA). For each sample, flagella were manually quantified in 25-30 images by a single unblinded operator.

### Phylogenetic tree and dendrogram construction

The genomes of all clinical isolates, ATCC 51730, ATCC 51783, and ATCC 51784 were annotated using Prokka 1.14.6[69] using default parameters. Roary 3.13.0[71] was applied to obtain the core genome using default settings. The core genome alignment file was used to generate a maximum-likelihood phylogenetic tree using the GTRGAMMA model and the rapid Bootstrap analysis options in RAxML 8.2.12[72,73]. The RaxML distances between the clinical isolates were used to generate an unrooted dendrogram based on the Neighbor joining method using the ape package in R. The transcriptome based dendrogram was generated similarly using the calculated Euclidean distances between the transcriptomes.

### Identification of isolate-specific enriched functional groups

Growth phase independent genes identified as differentially expressed in each isolate as compared to 2B3 were selected, and enrichment analysis of these DEGs was performed using topGO v2.40.0[68] with weight01 algorithm and Fisher exact test. The enriched functional groups identified in each of the isolates were plotted using Cytoscape[74]. Relations between Gene Ontology terms was derived from REVIGO[75].

### Reporting summary

Further information on research design is available in the Nature Portfolio Reporting Summary linked to this article.

## Data availability

All raw RNA-seq sequencing read files generated in this study have been submitted to SRA with Accession ID PRJNA752389 and GEO ID GSE181542. All base called long read and short read sequences for the ATCC genomes generated in this study have been submitted to ID

PRJNA625574. Additional raw data (including methylation data) generated in this study and R markup files have been uploaded to Zenodo with https://zenodo.org/records/13988484. Supplementary Data File 8 was also uploaded to Zenodo at the DOI above due to size (too large to upload to publication server).

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

## Acknowledgements

This work and all authors (S.G., C.J.W., A.G.M., A.L., L.N.H., B.T.H., E.R.F., J.H.Y., P.P.K.) were supported by the Division of Intramural Research of the National Institute of Allergy and Infectious Diseases (NIAID), NIH. This work utilized the computational resources of the NIH HPC Biowulf cluster (http://hpc.nih.gov). The content and views expressed in this work are those of the authors and do not necessarily represent the official views of the NIH or U.S. Government. We thank past and present members of the Bacterial Pathogenesis an Antimicrobial Resistance Section, NIAID, NIH and Dr. Anupama Khare for critical comments and discussion. We also thank the NIAID Research Technologies Branch at Rocky Mountain Laboratories in Montana, USA for their assistance with TEM, and Steven Holland for invaluable advice and feedback.

## Author contributions

S.G., C.J.W., A.L., P.P.K., and J.P.D. conceived of and designed the study and analyzes. J.P.D. obtained and managed funding for the study. C.J.W. generated RNA-seq libraries and J.H.Y. and P.P.K. sequenced them. S.G. and C.J.W. performed computational analysis of RNA-seq data and performed critical data management. A.G.M. and J.P.D. planned Nanopore methylome sequencing experiments and A.G.M. conducted Nanopore methylome sequencing and methylome data analysis. L.N.H., B.T.H., and E.R.F. performed TEM imaging of isolates. S.G. and C.J.W. generated the figures. J.P.D. supervised the study. S.G., C.J.W., P.P.K., and J.P.D. conducted critical review of both experimental data and computational analyzes. S.G., C.J.W., and J.P.D. wrote and revised the manuscript. All authors critically reviewed and/or edited the manuscript.

## Funding

## Competing interests

The authors declare no competing interests.
