## [Transparent Peer Review file · Nature Communications]

Transcriptional Diversification in a Human-Adapting Zoonotic Pathogen Drives Niche-Specific Evolution

Corresponding Author: Dr John Dekker

Version 0:

Reviewer comments:

Reviewer #1

(Remarks to the Author)

The manuscript proposed by Ghosh & Wu et al. to Nature Communications present a study very well done and informative. They made a clever separation of growth-phase dependent and independent gene regulations to clarified the RNAseq data. They present the evolution of a *B. hinzii* strain isolated at different time point from a unique patient. They explore the adaptation of the strain by RNAseq analyses of the gene regulation among isolates recovered during the infection period. They deciphered adaptation fo the bacteria to the host either in the vascular or GI compartments.

Here are my comments

Given the variability in growth rates among the 22 clinical isolates, would it be possible to get as supplemental figure the different growth rate for all the strains?

Does the strain growing faster or growing slower present statistical differences at 4h vs 10h ?

What are the plates used to isolate the bacterial colonies, are they LB? the same media used to grow them in liquid culture? can you implement the M&M section?

The initial OD to launch the culture is 0.01 but what are the OD at 4h and 10h for each strain when the RNA were recovered? What is the average time of generation of BH ? because you may have mixed at 4h maybe strain that already initiate growth and strains still under adaptation to liquid culture ?

In fig1 it will be informative to add the information of the time of collection of the different isolates during the 45-month infection in the immunocompromised host maybe as a small table or by implementing the strain names in 1b

Also on fig1b and F3b the names of the isolates from blood or the GI tract should be more highlighted or underlined to allow to discriminate them more than with only the "B" or "G" letter in the names (as well as in figure S3cd, S5abc, S6b and S7a)

Concerning the Biochemical pathway analysis of individual metabolic genes on fig 2 I suggest to add information relative to the Lof2FC of the gene involved in these pathways to visualized the real implication of these pathways in the change of metabolism. The differential gene expression may not be visualized gene per gene but colorization of the "number" (e.g. 1, 2, 3) of individual pathways like for a heat maps will allow to see which pathway are more regulated than others.

Line 229, 234 and 235 is it linear fold change or Log2FC ? I think it is Log2FC as it referred to fig 3b&c, I think it should be specified because $\text{Log}_2\text{FC}=9 = 512$ fold change

The authors show a correlation between flagellar gene downregulation and the capsular gene upregulation and link these observations to previously published work with the *risA* gene regulation in *B. pertussis*. Do the author looked at the *risA* gene integrity in their strains at the DNA level but also at the *risA* RNA production in the RNAseq datasets? Alternatively does flagellar gene downregulation and capsular gene upregulation are correlated with mutation in promoter regions driving these operon expressions? These could be added in the discussion section

On the Fig3b what is the rational of the order of the strains? why the blood or GI strains are not clustered together?

Does the correlation between flagellar gene downregulation and the capsular gene upregulation is also present with the 10h cultures ? a similar figure as figure 3 should be done as supplemental figure with the 10h cultures data without TEM and flagellated cells measuring.

On Fig3 panel d and e, on the figure or at least in the legend it should be labelled than the cells were cultured overnight to visualize and quantify the flagellation to not be confused with the 4h RNAseq data presented in panel b

The part about the chicken and human derived *B. hinzii* lineages should be really tempered as the 2 chicken isolates and

the 2 human isolates may not fully represent all the chicken and human isolates. What are the differences between the two chicken isolates? also what are the differences between the ATCC51730 and the 2B3 human strains ? I think that this part of the manuscript either should be removed or place at the very end after the analysis of evolutionary trajectories among the clinical lineages.

Reviewer #2

(Remarks to the Author)

• What are the noteworthy results?

The authors studied the transcriptome and the DNA methylation pattern of a considerable number of bacterial isolates (*Bordetella hinzii*) from the same host, an immunocompromised patient with IL12R deficiency. Several transcriptome modifications were observed in isolates cultivated in vitro, so we talk about transcriptome modification derived from changing in genome and epigenome, not related to a direct pathogen-host crosstalk. Results concerning the epigenomics are limited to the DNA methylation side. The identification of a DNA methylase and the corresponding target motif is interesting. It would be interesting to discuss also the presence of non-coding RNA, since the dataset the authors have comprise complete transcriptional data of the bacterium.

• Will the work be of significance to the field and related fields? How does it compare to the established literature? If the work is not original, please provide relevant references.

The work is original

• Does the work support the conclusions and claims, or is additional evidence needed?

The conclusions are supported, however the work may be more complete with an additional analysis of the non-coding RNA present in the samples. Many genomic mutations may find a correspondence on ncRNA regions and gain a stronger role in the regulation of gene expression.

• Are there any flaws in the data analysis, interpretation and conclusions? Do these prohibit publication or require revision?

• Is the methodology sound? Does the work meet the expected standards in your field?

The analysis of a transcriptome of an in-vitro culture of human isolates does not take into account the real transcriptomic changes happening during the pathogen-host interaction but can only provide a mirroring image of those genomic mutations involving coding regions. Transcriptomes changes in response to host signals, and this can only be observed by extracting RNA from bacterial cells directly isolated from the host.

• Is there enough detail provided in the methods for the work to be reproduced?

Data provided allows a good reproducibility of the experiments, however the time frame in which the samples were taken from the patient is unclear. Sampling has been described in <https://www.nature.com/articles/s41467-021-24668-7>, where a timeline of the isolate collection is shown. I suggest the authors to provide also in this manuscript a table with those information.

Supplementary data 8 is missing.

Specific comments:

Title: "Epigenomics" in the title is quite generic. Since only DNA methylation was investigated in this work, I would specify this. Change the title to be more precise (or improve the epigenomic analyses in the dataset). Maybe something like "Transcriptional and DNA methylation diversification in human adapted.." better reflect the aim of the study.

Line 47: the authors talk about "host immune evasion" but the host is immunocompromised, so it should be specified how the deficiency of IL12Rbeta1 influence the immunity response of the host and how it facilitates the pathogen immune evasion.

Line 69: specify "DNA methylation patterns"

Line 78: I would not connect mutation in DNA methyltransferase (mutations that involve the entire genome, in the points where that DNAmase act) with mutations on an associated recognized motif (that is a point mutation). The inheriting of the first mutation change the epigenetic landscape of the organism, the second act only on the related genes.

Line 182: "in other cases the transcriptome may be remarkably stable to large number of mutation" This is interesting.

Line 245: decrease flagellar production seems more likely happening in isolates from gut with respect to blood isolates.

Motility in the gut may be not so important as in liquid substrate, so this result is expected to me, however it is not discussed.

Line 263: Supplementary data 8 is missing.

Line 286: Were the motifs not methylated in 1B1a and 1B1b probably because the disruption of HHA2A:19560 DNAmase methylated in the other strains, where such gene was not stop gain-mutated?

Line 379-382: It is expected that transcriptomic changes in host-associated bacteria happens especially during the interaction with the host. In a liquid culture of bacteria DNA mutations can be observed, but host adaptation through epigenetic mechanisms cannot be fully observed. DNA methylation is only a part of the epigenetic mechanisms that the bacteria can develop to adapt in response to a changing environment and to develop virulence and pathogenicity traits. Did the authors investigate the non-coding RNA present in their dataset? Transcriptomic data provide a lot of reads that usually are filtered because they do not map on the reference genome but that should be interesting for analyses on the ncRNA side. The improvement of this kind of analyses would certainly complete the overview on which kind of non-transient epigenomic traits were changed during host adaptation. Several tools and database are available. The discussion section should anyway take into account the presence of such elements.

Line 383: The transcriptome changes the authors identify in the experimental conditions the closed are only a part of the transcriptomic changes that the bacteria can undergo during the infection. The in vitro growth of the isolates allows the evaluation of the DNA expression changes due to genome mutations, but the direct genome x genome interaction between host and pathogen can determine differential expression of the genes in the different moments of infection. Consider this in the discussion.

Line 430: Report here also the name of the DNA methylase gene and the motif recognizes, it can be useful in further investigation.

Discussion: the discussion section lacks in references (only few are provided).

Version 1:

Reviewer comments:

Reviewer #1

(Remarks to the Author)

Ghosh, Wu, et al. have raised all the critical comments and suggestions of both reviewers. The manuscript have been clarified and substantially improved, I have no other comment to make, the study can be published

Reviewer #2

(Remarks to the Author)

Thank you for responding all the questions/comments and for providing the additional study on non-coding RNAs.

I find exhaustive all the replies, and I agree with the authors' conclusion non to add the nc-RNA mapping in the manuscript.

The filtering of short molecules for sure impact the result.

I find the other modifications to text and addition of details make the manuscript more clear and the form more precise.

Response to Reviewers

“Transcriptional Diversification in a Human-Adapting Zoonotic Pathogen Drives Niche-Specific Evolution” by Ghosh, Wu, et al.

Manuscript NCOMMS-24-24597A

We sincerely thank the two reviewers for the thorough and thoughtful reviews of our manuscript. The critical comments and suggestions helped us to clarify and refine a number of points, and we believe these changes have substantially improved the manuscript. We are very highly appreciative of the effort and significant time donated to critique our work. Below we answer each of the questions raised. In particular, we (1) have provided detailed growth rate data and further analyzed the relationships between growth rates and transcriptomes at the 4 and 10 hour time points, (2) have performed analysis of ncRNA, and (3) have modified the text of the manuscript where appropriate to address the points the reviewers raise.

Reviewer #1 (Remarks to the Author):

Reviewer Comment: *The manuscript proposed by Ghosh & Wu et al. to Nature Communications present a study very well done and informative. They made a clever separation of growth-phase dependent and independent gene regulations to clarified the RNAseq data. They present the evolution of a B. hinzii strain isolated at different time point from a unique patient. They explore the adaptation of the strain by RNAseq analyses of the gene regulation among isolates recovered during the infection period. They deciphered adaptation fo the bacteria to the host either in the vascular or GI compartments.*

Author Response: We sincerely appreciate this reviewer’s highly insightful and detailed review of our manuscript and positive summary of our work above. We respond below to the comments individually, and indicate the modifications we have made, which we believe have significantly improved our manuscript.

Reviewer Comment: *Here are my comments Given the variability in growth rates among the 22 clinical isolates, would it be possible to get as supplemental figure the different growth rate for all the strains?*

Author Response: We thank the reviewer for raising this point. We have now included all growth curves as well as calculated doubling times as new parts to Supplementary Figure 1 (Figure S1b and S1c). To obtain growth curves, isolates were grown in LB medium at 37C overnight, followed by sub-culture in fresh LB medium at OD600 = 0.01. Growth was then monitored in a plate reader for 36 hrs. For each isolate, an early exponential phase interval was identified, and a linear function was fit to the log₂-transformed OD600 values in R. The slope of the fitted line was taken as a best estimate of growth rate, and the doubling time was taken as its inverse. In most cases the log-linear fit was excellent. In a couple cases, the slope was not mono-exponential. We believe the best single exponential log-linear fit is sufficient for the purposes of this analysis, so we did not employ more complicated multi-component models. Descriptive text has been included in the main manuscript (Lines 546-554).

Figure S1. Phylogenetic relationship between the isolates used in this study. a) Maximum likelihood phylogenetic tree of 22 clinical *B. hinzii* isolates and three ATCC strains constructed based on the core genome sequences of the isolates (see Methods for details). Clinical isolates with WT DnaQ and E9G DnaQ are indicated by red and black branches, respectively. The ATCC strains are indicated by blue branches. b) Growth curves of all isolates in LB medium. Data represent a total of 3 biological replicates, each with 4 technical replicates (n=12 measurements). The solid line represents mean values across all biological and technical replicates and standard deviation is represented with shading. The vertical lines indicate the interval selected for doubling time calculations. Blood isolates are indicated in blue and GI isolates are shown in black. c) Calculated doubling times of all isolates grown in LB medium. Pink bars indicate isolates with doubling times of 200 minutes or less (“fast” growth), green indicate doubling times between 200 and 300 minutes (“intermediate” growth), and blue indicates doubling time greater than 300 minutes (“slow” growth).

Reviewer Comment: *Does the strain growing faster or growing slower present statistical differences at 4h vs 10h ?*

Author Response: We thank the reviewer for raising this point. To explore this question, we made use of the GP-independent PCA plot calculated for all transcriptomes in Figure S2d (153 data points). We have reanalyzed these data and colored the data points based on the doubling time groups as calculated above and the time of harvest (4hr vs. 10hr). This new figure is included below and in the manuscript supplement as Figure S2e. The majority of the transcriptomes do not appear to demonstrate appreciable clustering by time of harvest or growth rate or combination of the two. We think this is consistent with the interpretation that our strategy to mitigate the effects of growth-phase dependence has been sufficiently successful. There are some outliers in this analysis (less than 10% of the total), the strongest of which correspond to 1G1 at 10hr. Isolate 1G1 contains a 0.5 megabase deletion in its genome, which substantially alters the transcriptome, and potentially the growth-dependence of different genes. We believe that this is likely related to the observed spread on the PCA plot associated with this isolate, though a detailed investigation into potentially altered growth-phase dependence in this single outlier isolate is really beyond the main points of this study, and we think that the main findings in the manuscript aren’t materially affected by the outlier behavior of isolate 1G1 in this respect.

Fig S2 (below). Identification of Growth phase dependent genes. a) Density plot of \log_2FC calculated between 4hr and 10hr timepoints in isolate 2B3 is shown. The distribution was fitted to a gaussian curve (blue) to identify cutoff at Z-score = ± 1 (dashed lines). Genes outside the cutoff were classified as growth phase dependent genes. b) PCA plot of the 153 transcriptomes based on all genes, colored by isolate (color scale at bottom right under “Isolates”). Circles indicate the 4hr time points and triangles indicate 10hr time points. c) PCA plot based on GP dependent genes, colored by isolate (color scale at bottom right under “Isolates”). Circles indicate the 4hr time points and triangles indicate 10hr time points. d) PCA plot for GP independent genes, colored by isolate (color scale at bottom right under “Isolates”). Circles indicate the 4hr time points and triangles indicate 10hr time points. e) PCA plot based on GP independent genes, colored based on their doubling time (color scale at bottom right under “Doubling time”). Circles indicate the 4hr time points and triangles indicate 10hr time points.

Reviewer Comment: What are the plates used to isolate the bacterial colonies, are they LB? the same media used to grow them in liquid culture? can you implement the M&M section?

Author Response: We thank the reviewer for pointing out this oversight. The isolates were recovered from frozen stocks on blood agar plates so that initial colonies could be visualized and confirmed prior to starting cultures for sequencing. Verified colonies from each plate were

inoculated into LB medium and grown overnight. The overnight culture was then aliquoted into fresh LB medium at OD 0.01 and grown for 4hr or 10hr before harvesting RNA. The intermediate overnight LB culture was employed to eliminate concerns about transcriptional adaptation associated with blood agar to LB media change. We have included these details in the manuscript and modified the M&M section accordingly (Lines 563-565).

Reviewer Comment: *The initial OD to launch the culture is 0.01 but what are the OD at 4h and 10h for each strain when the RNA were recovered? What is the average time of generation of BH ? because you may have mixed at 4h maybe strain that already initiate growth and strains still under adaptation to liquid culture ?*

Author Response: We thank the reviewer for raising this point. We have now added the OD600 values for each of the cultures at the times of RNA harvest in Supplemental Data 1. We have also calculated and added the doubling times of all the isolates in Figure S1c as noted above. The isolates were grown in an LB culture overnight (from blood agar) before they were sub-cultured the LB at 0.01 and grown for RNA harvest. We believe this protocol should eliminate liquid media adaptation effects. We are aware we are harvesting RNA from isolates that are at different points in the growth curve as a result of the different growth rates, and this was the purpose of the work to identify the growth phase-dependent and growth-phase independent genes. The analysis of the manuscript focuses only on the genes identified as growth phase-independent (with all the acknowledged caveats).

Reviewer Comment: *In fig1 it will be informative to add the information of the time of collection of the different isolates during the 45-month infection in the immunocompromised host maybe as a small table or by implementing the strain names in 1b Also on fig1b and F3b the names of the isolates from blood or the GI tract should be more highlighted or underlined to allow to discriminate them more than with only the “B” or “G” letter in the names (as well as in figure S3cd, S5abc, S6b and S7a)*

Author Response: We agree fully with the reviewer’s suggestions. We have modified Figure 1 to include the timeline of isolate collection at the top. In addition, we have modified all other relevant figures (including S3, S5, S6, and S7) throughout the manuscript to indicate isolates derived from blood and GI with different colors.

Fig 1 (below). *B. hinzii* isolates evolved extensive transcriptomic diversity during prolonged infection of human host. a) Timeline of isolate collection. Isolates were numbered by the day of collection, followed by culture source (Blood or Gastrointestinal), followed by number and letter designating individual morphotypes. Blood isolates are indicated in blue and GI isolates are indicated in black. b) Schematic illustration of experimental workflow for transcriptome analysis. Twenty-two *B. hinzii* isolates were cultured overnight in triplicate, followed by sub-culturing, RNA harvest at 4hr and 10hr and Illumina library preparation. The sequencing reads were trimmed, filtered, and mapped to reference isolate 2B3, followed by comparison of normalized counts of growth phase independent genes to identify differentially expressed genes using DESeq2. c) Clustered heat map of relative Euclidean distances between the transcriptomes (only growth phase independent genes) of all isolate pairs, except 3B1 at the 4hr timepoint. The color of each cell corresponds to the Euclidean distance between isolate pairs on a scale of 0 -

80, represented in the color band at right. d) Neighbor joining tree based on the pairwise distances between isolate genomes. e) Neighbor joining tree based on Euclidean distances between isolate transcriptomes. f) Correlation plot between the genotypic distance and transcriptome distance (Euclidean distance) between pairs of isolates at 4hr time point. DnaQ WT isolates are shown in red; isolates in black are DnaQ E9G proof-reading deficient hypermutators, and the isolate shown in blue is 3B1, which is a compound hypermutator and an outlier. Linear fit is to all points including 3B1, which are shown as blue dots.

Reviewer Comment: Concerning the Biochemical pathway analysis of individual metabolic genes on fig 2 I suggest to add information relative to the Lof2FC of the gene involved in these pathways to visualize the real implication of these pathways in the change of metabolism. The

differential gene expression may not be visualized gene per gene but colorization of the “number” (e.g. 1, 2, 3) of individual pathways like for a heat maps will allow to see which pathway are more regulated than others.

Author Response: We agree with the reviewer and think this is an excellent suggestion. We have now colored the individual arrows based on the \log_2FC values of these enzymes between Blood and GI isolates based on group DESeq2 calculations. The color scale bar is included in the figure.

Fig 2. Niche specific differential regulation of central metabolic pathways. A total of 268 differentially expressed genes were identified between isolates harvested from blood and the GI tract, of which 25 corresponded to metabolic genes in the KEGG database. The figure indicates the enzyme, intermediates, and products for each inferred reaction, and enzymes are numbered as defined below. Arrow color defined in key indicates the \log_2FC values of differential expression based on group DESeq2 calculations. Enzyme labels are as follows: 1) nitronate

monooxygenase, 2) nitrite reductase (NADH) large subunit, 3) nitrite reductase (NADH) small subunit, 4) 4-methoxybenzoate monooxygenase, 5) dihydropyrimidine dehydrogenase subunit A, 6) UDP-N-acetyl-D-glucosamine 6-dehydrogenase, 7) acylamidase, 8) catalase-peroxidase, 9) 3-hydroxycinnamic acid hydroxylase, 10) rhodocoxin reductase, 11) cystathionine beta-lyase, 12) dihydrolipoyl dehydrogenase 13) pyruvate dehydrogenase E2 component, 14) aconitate hydratase, 15) glutamate synthase (NADPH) large chain, 16) glutamate synthase (NADPH) small chain, 17) 4-hydroxy-tetrahydrodipicolinate synthase, 18) diaminobutyrate--2-oxoglutarate transaminase, 19) L-2,4-diaminobutyric acid acetyltransferase, 20) L-ectoine synthase, 21) ectoine hydroxylase, 22) maleamate amidohydrolase, 23) NADH-dependent formate dehydrogenase delta subunit FdsD, 24) aspartyl-tRNA(Asn)/glutamyl-tRNA(Gln) amidotransferase subunit A.

Reviewer Comment: Line 229, 234 and 235 is it linear fold change or Log2FC ? I think it is Log2FC as it referred to fig 3b&c, I think it should be specified because $\text{Log}_2\text{FC}=9 = 512$ fold change

Author Response: We thank the reviewer for pointing this lack of clarity. The values are supposed to be $\log_2\text{FC}$ in the text (and we appreciate that some of the changers are large). We have modified the manuscript to clarify in these places with “32-fold” and “512-fold”.

Reviewer Comment: The authors show a correlation between flagellar gene downregulation and the capsular gene upregulation and link these observations to previously published work with the *risA* gene regulation in *B. pertussis*. Do the author looked at the *risA* gene integrity in their strains at the DNA level but also at the *risA* RNA production in the RNAseq datasets?

Author Response: The genes encoding RisA and RisS appear to have intact coding regions within the isolates. As these genes have not been studied in *B. hinzii*, there is not a defined “functional” canonical sequence to which we can align our sequences, and there is divergence from the studied *B. pertussis* genes. Thus, we cannot know for sure without additional experimental work whether RisA and RisS are functional within the *B. hinzii* species. Among the patient isolates, RisA sequences were identical among all isolates, while RisS had 5 different substitutions, each in different groups of isolates:

Mutations	Isolates
Y93S	3G1, 3G2
R154L	1B1a, 1B1b
V178E	1G2,2G2, 3G1
R234L	3B1
D237Y	all DnaQ E9G isolates

The D237Y substitution had been discussed in the earlier complementary publication (Launay et al, *Nat Commun*, 2021) and is suggestive of ancestral substitution in the DnaQ E9G lineage. The genes encoding RisA and RisS are both expressed in all 22 isolates. The $\log_2\text{FC}$ gene expression values for *risA* were not significantly different in the isolates when compared with “ancestral” 2B3. (Significance was defined by at least a two-fold change in expression in

addition to a significant adjusted p value.) For the *risS* gene, upregulation with log₂FC values of 1.22 and 1.0015 were observed in isolates 1B1b and 2B2b, respectively.

	risS	risA
1B1a	0.00	0.00
1B1b	1.22	0.00
1G1	0.00	0.00
1G2	0.00	0.00
2B1	0.00	0.00
2B2a	0.00	0.00
2B2b	1.00	0.00
2G1	0.00	0.00
2G2	0.00	0.00
3B1	0.00	0.00
3G1	0.00	0.00
3G2	0.00	0.00
3G3	0.00	0.00
4G1a	0.00	0.00
4G1b	0.00	0.00
4G1c	0.00	0.00
4G1d	0.00	0.00
5G1a	0.00	0.00
5G1b	0.00	0.00
7B1	0.00	0.00
8B1	0.00	0.00

While we think studies of the RisA/S system would likely be very informative, we feel that experimental work to confirm whether the RisA/S system is functional in these isolates, and whether any of the above mutations alter function is beyond the scope of the current work. We have added additional explanatory text in lines 263-266 and 488-492.

Reviewer Comment: *Alternatively does flagellar gene downregulation and capsular gene upregulation are correlated with mutation in promoter regions driving these operon expressions? These could be added in the discussion section*

Author Response: This is a great question, and we spent a lot of time analyzing prior to writing the manuscript. We do not yet understand the molecular mechanism of flagellar gene downregulation in each of the isolates and we were unable to discover a straightforward correlation between a single mutation, or single group of co-occurring mutations, or different mutations in a single gene. It appears likely to us that downregulation evolved multiple times (at least twice, possibly more) convergently with different groups of mutations in different isolates. Given the large number of mutations and groups of mutations that are shared in different combinations between these isolates, and without knowledge of the mechanism of downregulation, it is difficult to make a rigorous and supported statement about which mutations

are causative. This was the reason we did not comment further in the first version of the manuscript. In this revised version, we have added additional cautious interpretive text in the manuscript in lines 472-485.

Reviewer Comment: *On the Fig3b what is the rationale of the order of the strains? why the blood or GI strains are not clustered together?*

Author Response: The isolates were ordered based on the day of isolation. We agree with the reviewer that grouping the blood and GI isolates together is a more logical way to order them, and the figure has now been reordered to segregate blood and GI isolates.

Reviewer Comment: *Does the correlation between flagellar gene downregulation and the capsular gene upregulation is also present with the 10h cultures ? a similar figure as figure 3 should be done as supplemental figure with the 10h cultures data without TEM and flagellated cells measuring.*

Author Response: Figure 3b shows the differential expression of genes in all isolates with respect to 2B3. In contrast to the per-isolate counts data, which are presented separately for 4hr and 10hr datasets, differential expression is calculated using a DESeq2 multifactorial design that explicitly takes into account 4hr and 10hr grouping with time as an independent variable. (This is described in the Methods.) We employed this method for the differential gene expression calculations to mitigate residual time-dependent bias that remained after removing the strongly growth-phase dependent genes. Thus, the differential expression map incorporates both 4hr and 10hr data. The heat map on the right in Figure 3b represents the rlog counts of these genes in 2B3 at 4hr and 10hr and provides a reference.

Reviewer Comment: *On Fig3 panel d and e, on the figure or at least in the legend it should be labelled that the cells were cultured overnight to visualize and quantify the flagellation to not be confused with the 4h RNAseq data presented in panel b.*

Author Response: We thank the reviewer for this suggestion, which is an important clarification. We have now included this information in the figure legend (and full details are included in the Methods section).

Reviewer Comment: *The part about the chicken and human derived B. hinzii lineages should be really tempered as the 2 chicken isolates and the 2 human isolates may not fully represent all the chicken and human isolates. What are the differences between the two chicken isolates? also what are the differences between the ATCC51730 and the 2B3 human strains ? I think that this part of the manuscript either should be removed or place at the very end after the analysis of evolutionary trajectories among the clinical lineages.*

Author Response: We agree fully with the reviewer's concern and suggestion. On further careful consideration, we have chosen to remove this part of the manuscript, including Figure S7, and Supplemental Data File 10, and hope to have more data in the future to support a more complete and rigorous comparison of chicken and human-adapted isolates.

Reviewer #2 (Remarks to the Author):

Reviewer Comment:

• *What are the noteworthy results?*

The authors studied the transcriptome and the DNA methylation pattern of a considerable number of bacterial isolates (Bordetella hinzii) from the same host, an immunocompromised patient with IL12R deficiency. Several transcriptome modifications were observed in isolates cultivated in vitro, so we talk about transcriptome modification derived from changing in genome and epigenome, not related to a direct pathogen-host crosstalk. Results concerning the epigenomics are limited to the DNA methylation side. The identification of a DNA methylase and the corresponding target motif is interesting.

Author Response: We sincerely appreciate this reviewer's detailed review and the highly valuable insights and suggestions. We believe the modifications we have made in response have significantly improved our manuscript.

Reviewer Comment: *It would be interesting to discuss also the presence of non-coding RNA, since the dataset the authors have comprise complete transcriptional data of the bacterium.*

Author Response: We agree with the reviewer that a rigorous analysis of non-coding RNA could add significantly to the study and is missing. Below we describe this analysis, but there is a critical caveat: The RNA extraction methods were not designed to capture RNAs less than 200 nucleotides in length. We expect smaller RNAs to have been eluted significantly and not represented in the libraries in a consistent manner. We did not include any internal standards to know how the captured molar concentrations and normalized read counts in this size range relate to input values. This is a critical limitation, and while we performed the analysis below, we have serious concerns about its rigor and validity, and do not feel that we can include the results in the manuscript.

We predicted small RNAs (sRNAs) using the ANNOgesic approach (<https://annogesic.readthedocs.io/en/latest/>) with default parameters. The software relies on coverage information to define transcript boundaries and identifies potential regulatory regions. To predict the base set of sRNAs, we used the RNA-seq datasets from isolate 2B3, which was used as the reference isolate for the other analyses. We utilized the datasets with 150 paired-end reads (as opposed to 75 single-end) for this analysis to increase the accuracy of the analysis performed by the ANNOgesic pipeline. A total of 38 sRNAs were predicted by ANNOgesic, of which 36 were antisense RNA and had a predicted coding region that could be regulated, and two were present in predicted intergenic regions. In addition, a total of 74 tRNAs and 2 ncRNAs were also identified based on NCBI PGAP annotation. We combined all of the above identified species of RNA (n = 112) which we refer to collectively as “ncRNAs” in the analysis.

To examine expression of the ncRNAs, we performed alignment of all reads (75bp SE) for the 153 transcriptomes to the 2B3 reference genome using BWA, followed by determination of counts for each ncRNA using HTSeq, using the same parameters as were used as the mRNA analysis as described in Methods. ncRNA with fewer than 10 counts in all samples were not included in the analysis. We then calculated the rlog counts and differential expression of ncRNA between the Blood and GI groups (Fig. RR1). Of the 112 sRNAs tested, 3 sRNA showed significant differential expression. As noted above, we have serious concerns about its rigor and validity of this analysis due to the lack of confidence in how reliably the small RNAs were captured, and do not feel that we can interpret the results further or include the results in the manuscript.

Fig RR1. Analysis of ncRNA. a) ncRNA rlog count values for all isolates at 4hr and 10hr time points. b) Volcano plot displaying differentially expressed genes (all RNAs including mRNAs and ncRNAs) between Blood and GI isolate groups. 3/112 ncRNAs met thresholds for significance, all of which were anti-sense RNAs. These are highlighted in cyan.

Reviewer Comment:

• *Will the work be of significance to the field and related fields? How does it compare to the established literature? If the work is not original, please provide relevant references.*

The work is original

• *Does the work support the conclusions and claims, or is additional evidence needed?*

The conclusions are supported, however the work may be more complete with an additional analysis of the non-coding RNA present in the samples. Many genomic mutations may find a correspondence on ncRNA regions and gain a stronger role in the regulation of gene expression.

Author Response: As noted above, we agree fully with the reviewer that a rigorous analysis of non-coding RNA could add significantly to the study, but given the considerations of RNA extraction approach, we do not feel confident in the analysis above to include in the manuscript, as indicated above.

Reviewer Comment:

• *Are there any flaws in the data analysis, interpretation and conclusions? Do these prohibit publication or require revision?*

• *Is the methodology sound? Does the work meet the expected standards in your field?*

The analysis of a transcriptome of an in-vitro culture of human isolates does not take into account the real transcriptomic changes happening during the pathogen-host interaction but can only provide a mirroring image of those genomic mutations involving coding regions.

Transcriptomes changes in response to host signals, and this can only be observed by extracting RNA from bacterial cells directly isolated from the host.

Author Response: We agree fully with the reviewer on this point. We would ultimately like to understand gene expression within these host adapted isolates in the native context in which they evolved. We believe that some of this evolution likely occurred intracellularly within host phagocytes. The current analysis was a starting point and we are planning future steps that will involve dual RNA-seq with macrophages obtained from the patient in combination with the various lineages. This is beyond the scope of the current analysis, but we believe the data obtained in this work would be foundational for further studies. We have added discussion of this point as a caveat to the manuscript in Lines 519-524 to address the reviewer's point:

“An important caveat of this work is that transcriptomes were examined in standardized LB culture media. It is likely that there are transcriptional processes that are dynamically regulated by the native host environment that may have undergone specific selection and adaptation that were not appreciated in this study. Future work aiming at transcriptome characterization in native contexts, including intracellular vesicles of phagocytes will be necessary to answer these questions.”

Reviewer Comment:

• *Is there enough detail provided in the methods for the work to be reproduced?*
Data provided allows a good reproducibility of the experiments, however the time frame in which the samples were taken from the patient is unclear. Sampling has been described in <https://www.nature.com/articles/s41467-021-24668-7>, where a timeline of the isolate collection is shown. I suggest the authors to provide also in this manuscript a table with those information.

Author Response: We thank the reviewer for this suggestion and have now included a timeline as Figure 1a. (Figure 1 is included above in response to the first reviewer's question.)

Reviewer Comment: *Supplementary data 8 is missing.*

Author Response: This file was too large to upload to the Nature web portal, so we had uploaded it to the Zenodo server. This is indicated in the Data Availability paragraph, but we appreciate this may not have been clear. We have reworded and simplified the Data Availability paragraph and emphasized that this file is found by using the Zenodo DOI.

Reviewer Comment:

Specific comments:

Title: “Epigenomics” in the title is quite generic. Since only DNA methylation was investigated in this work, I would specify this. Change the title to be more precise (or improve the epigenomic analyses in the dataset). Maybe something like “Transcriptional and DNA methylation diversification in human adapted..” better reflect the aim of the study.

Author Response: We agree with this reviewer and think the best solution may be just to remove “epigenomic” from the title to yield “Transcriptional Diversification in a Human-Adapting Zoonotic Pathogen Drives Niche-Specific Evolution”.

Reviewer Comment: *Line 47: the authors talk about “host immune evasion” but the host is immunocompromised, so it should be specified how the deficiency of IL12Rbeta1 influence the immunity response of the host and how it facilitates the pathogen immune evasion.*

Author Response: We thank the reviewer for raising this point. While this patient had impaired IL-12Rβ1-dependent intracellular immunity, it is expected that components of innate pattern-recognition immunity would be intact to mediate such immunologic selection. The abstract is very word-constrained, so we have expanded on this point in the main text in Lines 467-469.

Reviewer Comment: *Line 69: specify “DNA methylation patterns”*

Author Response: We thank the reviewer for identifying this lack of precision, and we have modified as suggested.

Reviewer Comment: *Line 78: I would not connect mutation in DNA methyltransferase (mutations that involve the entire genome, in the points where that DNAmase act) with mutations on an associated recognized motif (that is a point mutation). The inheriting of the first mutation change the epigenetic landscape of the organism, the second act only on the related genes.*

Author Response: We agree with this point and have reworded to: “Very little is known about how mutations in DNA methyltransferases modify transcription during host adaptation.”

Reviewer Comment: *Line 182: “in other cases the transcriptome may be remarkably stable to large number of mutation” This is interesting. Line 245: decrease flagellar production seems more likely happening in isolates from gut with respect to blood isolates. Motility in the gut may be not so important as in liquid substrate, so this result is expected to me, however it is not discussed.*

Author Response: We agree that loss of motility (and activation of inversely related capsule/biofilm synthesis) may have different consequences and undergo different selection depending on environmental medium and viscosity. However, in the original version of Figure 3, where the GI and blood isolates were not well-segregated, it may have been difficult to appreciate that 9/13 (69%) of the GI isolates have strongly downregulated flagellar genes, compared with 5/9 (56%) of the blood isolates (note this includes the reference ancestor 2B3). Thus we are not sure there is a meaningful difference in occurrence in this small set. The new Figure 3 (above, response to reviewer 1) hopefully makes the distinction between GI (black, left) and blood (blue, right) isolates clearer. We have also reworded the unclear text in this section (lines 257-258).

Reviewer Comment: *Line 263: Supplementary data 8 is missing.*

Author Response: As noted above, this file was too large to upload to the Nature web portal, so we had uploaded it to the Zenodo server. This is indicated this in the Data Availability paragraph, which we have rewritten to make this point more clear.

Reviewer Comment: *Line 286: Were the motifs not methylated in 1B1a and 1B1b probably because the disruption of HHA2A:19560 DNAmase methylated in the other strains, where such gene was not stop gain-mutated?*

Author Response: Yes, this is what we believe to be the case. Given (1) that only one Type III methyltransferase was annotated, corresponding to the Type III motif, and that (2) this gene has stop-gain mutations only in 1B1a/b that are missing methylation of this Type III motif, we feel this is very strong evidence, even without a complementation experiment. We have modified the text to make this clearer:

“Given the exact correspondence between the stop gain mutation in the Type III DNA methyltransferase shared by 1B1a and 1B1b and the loss of the Type III methylated motif in these same isolates, we concluded that it is highly likely this enzyme is responsible for methylation at this position, and that the stop gain mutation was responsible for its loss.”

Reviewer Comment: *Line 379-382: It is expected that transcriptomic changes in host-associated bacteria happens especially during the interaction with the host. In a liquid culture of bacteria DNA mutations can be observed, but host adaptation through epigenetic mechanisms cannot be fully observed. DNA methylation is only a part of the epigenetic mechanisms that the bacteria can develop to adapt in response to a changing environment and to develop virulence and pathogenicity traits.*

Author Response: As noted above, we agree fully with the reviewer on this point. We would ultimately like to understand gene expression within these host adapted isolates in the native context in which they evolved. We believe that some of this evolution likely occurred intracellularly within host phagocytes. The current analysis was a starting point and we are planning future steps that will involve dual RNA-seq with macrophages obtained from the patient in combination with the various lineages. This is beyond the scope of the current analysis, but we believe the data obtained in this work would be foundational for further studies. We have added discussion of this point as a caveat to the manuscript in Lines 519-524 to address the reviewer’s point.

Reviewer Comment: *Did the authors investigate the non-coding RNA present in their dataset? Transcriptomic data provide a lot of reads that usually are filtered because they do not map on the reference genome but that should be interesting for analyses on the ncRNA side. The improvement of this kind of analyses would certainly complete the overview on which kind of non-transient epigenomic traits were changed during host adaptation. Several tools and database are available. The discussion section should anyway take into account the presence of such elements.*

Author Response: Please see full response above with analysis of ncRNAs.

Reviewer Comment: *Line 383: The transcriptome changes the authors identify in the experimental conditions the chosed are only a part of the transcriptomic changes that the bacteria can undergo during the infection. The in vitro growth of the isolates allows the evaluation of the DNA expression changes due to genome mutations, but the direct genome x genome interaction between host and pathogen can determine differential expression of the genes in the different moments of infection. Consider this in the discussion.*

Author Response: As explained in more detail in the two responses above, we agree fully with the reviewer on this point and we have added discussion to the manuscript in Lines 519-524 to address the reviewer's point.

Reviewer Comment: *Line 430: Report here also the name of the DNA methylase gene and the motif recognizes, it can be useful in further infestigation.*

Author Response: The DNA Methyltransferase gene was identified through the use of an HMM profile model we had previously developed and published (Tisza et al, *Nat Commun.* 2023 Jul 10;14(1):4082). Thus, there is not a formal "name" for this methyltransferase. The full annotated sequence of the methyltransferase is included with the uploaded genome assemblies and the motif recognized is indicated in the text (AGCG4mCCY).

Reviewer Comment: *Discussion: the discussion section lacks in references (only few are provided).*

Author Response: We thank the reviewer for this comment and have added additional references to the discussion.